# `BYOL-Explore:`
# Exploration by Bootstrapped Prediction

**Zhaohan Daniel Guo**[*]
DeepMind
danielguo@deepmind.com

**Shantanu Thakoor**[*]
DeepMind

**Miruna Pîslar**[*]
DeepMind

**Bernardo Avila Pires**[*]
DeepMind

**Florent Altché**[*]
DeepMind

**Corentin Tallec**[*]
DeepMind

**Alaa Saade**
DeepMind

**Daniele Calandriello**
DeepMind

**Jean-Bastien Grill**
DeepMind

**Yunhao Tang**
DeepMind

**Michal Valko**
DeepMind

**Rémi Munos**
DeepMind

**Mohammad Gheshlaghi Azar**[*]
DeepMind
mazar@deepmind.com

**Bilal Piot**[*]
DeepMind
piot@deepmind.com

## Abstract

We present `BYOL-Explore`, a conceptually simple yet general approach for curiosity-driven exploration in visually-complex environments. `BYOL-Explore` learns a world representation, the world dynamics, and an exploration policy all-together by optimizing a single prediction loss in the latent space with no additional auxiliary objective. We show that `BYOL-Explore` is effective in `DM-HARD-8`, a challenging partially-observable continuous-action hard-exploration benchmark with visually-rich 3-D environments. On this benchmark, we solve the majority of the tasks purely through augmenting the extrinsic reward with `BYOL-Explore`'s intrinsic reward, whereas prior work could only get off the ground with human demonstrations. As further evidence of the generality of `BYOL-Explore`, we show that it achieves superhuman performance on the ten hardest exploration games in `Atari` while having a much simpler design than other competitive agents.

## 1 Introduction

Exploration is essential to *reinforcement learning* (RL) [67], especially when extrinsic rewards are sparse or hard to reach. In rich environments, the variety of meaningful directions of exploration makes it impractical to visit everything. Thus, the question becomes: how can an agent determine which parts of the environment are interesting to explore? One promising paradigm to address this challenge is curiosity-driven exploration. It consists of (i) learning a predictive model of some information about the world, called a *world model*, and (ii) using discrepancies between predictions of the world model and real experience to build intrinsic rewards [59, 66, 60, 34, 51, 52, 2]. An RL agent optimizing these intrinsic rewards drives itself towards states where the world model is incorrect or imperfect, generating new trajectories on which the world model can be improved. In other words, the properties of the world model influence the quality of the exploration policy, which in turn gathers new data to shape the world model itself. Thus, it can be important not to treat learning

---

[*]Equal contribution.

the world model and learning the exploratory policy as two separate problems, but instead altogether as a single joint problem to solve.

In this paper, we present `BYOL-Explore`, a curiosity-driven exploration algorithm whose appeal resides in its conceptual simplicity, generality, and high performance. `BYOL-Explore` learns a world model with a self-supervised prediction loss, and uses the same loss to train a curiosity-driven policy, thus using a single learning objective to solve both the problem of building the world model's representation and the curiosity-driven policy. Our approach builds upon *Bootstrap Your Own Latent* (`BYOL`), a latent-predictive self-supervised method which predicts an older copy of its own latent representation. This bootstrapping mechanism has already been successfully applied in computer vision [20, 56], graph representation learning [71], and representation learning in RL [24, 62]. However, the latter works focus primarily on using the world-model for representation learning in RL whereas `BYOL-Explore` takes this one step further, and not only learns a versatile world model but also uses the world model's loss to drive exploration.

We evaluate `BYOL-Explore` on `DM-HARD-8` [22], a suite of 8 complex first-person-view 3-D tasks with sparse rewards. These tasks demand efficient exploration since in order to reach the final goal and obtain the reward they require completing a sequence of precise, orderly interactions with the physical objects in the environment, unlikely to happen under a vanilla random exploration strategy (see Fig. 2 and the videos in supplementary materials). To show the generality of our method we also evaluate `BYOL-Explore` on the ten hardest exploration `Atari` games [5]. In all these domains, `BYOL-Explore` outperforms other prominent curiosity-driven exploration methods, such as *Random Network Distillation* (`RND`) [8] and *Intrinsic Curiosity Module* (`ICM`) [51]. In `DM-HARD-8`, `BYOL-Explore` achieves human-level performance in the majority of the tasks using only the extrinsic reward augmented with `BYOL-Explore`'s intrinsic reward, whereas previously significant progress required human demonstrations [22]. Remarkably, `BYOL-Explore` achieves this performance using only a single world model and a single policy network concurrently trained across all tasks. Finally, as further evidence of its generality, `BYOL-Explore` achieves superhuman performance in the ten hardest exploration `Atari` games [5] while having a simpler design than other competitive agents, such as `Agent57` [3, 4] and `Go-Explore` [14, 15].[2]

## 2   Related Work

There is a large body of research in building world models either for planning [66, 63, 27, 26, 61], representation learning [62, 24, 41, 19] or curiosity-driven exploration [59, 68, 60, 34, 51, 52, 2, 63, 21, 65]. Most works consider world models that predict the entire observations [58, 48, 16, 19], which necessitates a loss in pixel space when observations are visually complex images. Some works have considered predicting latent representations, whether they are random projections [7, 8], or learned representations from a separate model, such as an inverse dynamics model [51] or an auto-encoder [25, 7]. Finally, some RL works [61] have focused on predicting lower-dimensional quantities such as the extrinsic reward, the action-selection policy, and the value function to build a world model.

Our `BYOL-Explore`'s world model operates in latent space and uses the same loss both for representation and intrinsic reward, simplifying and unifying representation learning and exploration. `BYOL-Explore`'s world model is derived from recent self-supervised representation learning methods [20, 56, 55, 71] and is similar to the ones in self-supervised RL [62, 24]. These previous works focused on the benefit of shaping representations for policy learning and have not looked into exploration. We build on this previous work to show that we can take the impact of a good representation technique further and use it to drive exploration.

While our approach belongs to the curiosity-driven exploration paradigm [50, 42, 49, 59, 5, 68, 60, 34, 51, 52, 2, 63], other exploration paradigms have also been proposed. The maximum entropy paradigms try to steer the agent to a desired distribution of states (or state-action pairs) that maximizes the entropy of visited states [29, 69, 70, 23]. The goal-conditioned paradigm has the agent set its own goal to drive exploration [57, 1, 17, 75, 47, 12, 82, 28, 15, 54, 80, 53]. The reward-free exploration

---

[2]Contrary to `Agent57`, `BYOL-Explore` neither requires episodic memory nor using an additional bandit mechanism to mix long-term and short-term rewards. As opposed to `Go-Explore`, we do not have to explicitly keep in memory a set of diverse goal-states to visit, which requires setting additional hyper-parameters that are environment-dependent.

paradigm consists of training an agent to explore the environment such that it would be able to produce a near-optimal policy for *any* possible reward function [37, 39, 45, 78, 74, 9, 79, 81].

# 3    Method

Our agent has three components:   a self-supervised latent-predictive world-model called `BYOL-Explore`, a generic reward normalization and prioritization scheme, and an off-the-shelf RL agent that can optionally share its own representation with `BYOL-Explore`'s world model.

## 3.1    Background and Notation

We consider a discrete-time interaction process [44, 35, 36, 13] between an agent and its environment where, at each time step $t \in \mathbb{N}$, the agent receives an observation $o_t \in \mathcal{O}$ and generates an action $a_t \in \mathcal{A}$. We consider an environment with stochastic dynamics $p : \mathcal{H} \times \mathcal{A} \to \Delta_{\mathcal{O}}^3$ that maps a history of past observations-actions and a current action to a probability distribution over future observations. More precisely, the space of past observations-actions is $\mathcal{H} = \bigcup_{t \in \mathbb{N}} \mathcal{H}_t$ where $\mathcal{H}_0 = \mathcal{O}$ and $\forall t \in \mathbb{N}^*, \mathcal{H}_{t+1} = \mathcal{H}_t \times \mathcal{A} \times \mathcal{O}$. We consider policies $\pi : \mathcal{H} \to \Delta_{\mathcal{A}}$ that maps a history of past observations-actions to a probability distribution over actions. Finally, an extrinsic reward function $r_e : \mathcal{H} \times \mathcal{A} \to \mathbb{R}$ maps a history of past observations-actions to a real number.

## 3.2    Latent-Predictive World Model

`BYOL-Explore` world model is a multi-step predictive world model operating at the latent level. It is inspired by the self-supervised learning method `BYOL` in computer vision and adapted to interactive environments (see Section 3.1). Similar to `BYOL`, `BYOL-Explore` model trains an online network using targets generated by an exponential moving average (EMA) target network. However, `BYOL` obtains its targets by applying different augmentations to the same observation as the online representation, whereas `BYOL-Explore` model gets its targets from future observations processed by an EMA of the online network, with no hand-crafted augmentation. Also `BYOL-Explore` model, uses a recurrent neural network (RNN) [33, 11] to build the agent state, i.e., the state of RNN, from the history of observations, whereas the original `BYOL` only uses a feed-forward network for encoding the observations. In the remainder of this section, we will explain: (i) how the online network builds future predictions, (ii) how targets for our predictions are obtained through a target network, (iii) the loss used to train the online network, and (iv) how we compute the uncertainties of the world model.

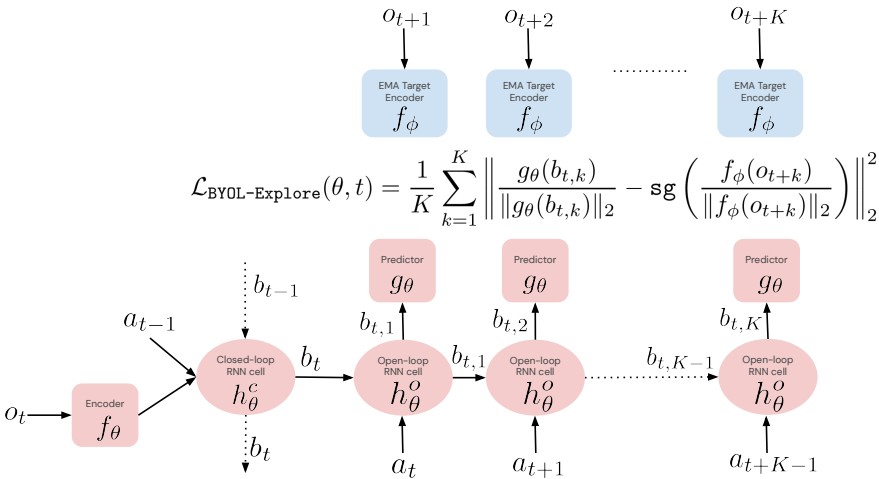

$$\mathcal{L}_{\texttt{BYOL-Explore}}(\theta, t) = \frac{1}{K} \sum_{k=1}^{K} \left\| \frac{g_\theta(b_{t,k})}{\|g_\theta(b_{t,k})\|_2} - \texttt{sg}\left( \frac{f_\phi(o_{t+k})}{\|f_\phi(o_{t+k})\|_2} \right) \right\|_2^2$$

Figure 1: `BYOL-Explore`'s Neural Architecture (see main text for details).

---

[3] We write $\Delta_{\mathcal{Y}}$ the set of probability distributions over a set $\mathcal{Y}$.

**(i) Future Predictions.** The online network is composed of an encoder $f_\theta$ that transforms an observation $o_t$ into an observation-representation $f_\theta(o_t) \in \mathbb{R}^N$, where $N \in \mathbb{N}^*$ is the embedding size. The observation-representation $f_\theta(o_t)$ is then fed alongside the previous action $a_{t-1}$ to a RNN cell $h_\theta^c$ that is referred as the close-loop RNN cell. It computes a representation $b_t \in \mathbb{R}^M$ of the history $h_t \in \mathcal{H}_t$ seen so far as $b_t = h_\theta^c(b_{t-1}, a_{t-1}, f_\theta(o_t))$, where $M \in \mathbb{N}^*$ is the size of the history-representation. Then, the history-representation $b_t$ is used to initialize an open-loop RNN cell $h_\theta^o$ that outputs open-loop representations $(b_{t,k} \in \mathbb{R}^M)_{k=1}^{K-1}$ as $b_{t,k} = h_\theta^o(b_{t,k-1}, a_{t+k-1})$ where $b_{t,0} = b_t$ and $K$ is the open-loop horizon. The role of the open-loop RNN cell is to *simulate* future history-representations while observing only the future actions. Finally, the open-loop representation $b_{t,k}$ is fed to a predictor $g_\theta$ to output the open-loop prediction $g_\theta(b_{t,k}) \in \mathbb{R}^N$ at time $t+k$ that plays the role of our future prediction at time $t+k$.

**(ii) Targets and Target Network.** The target network is an observation encoder $f_\phi$ whose parameters are an EMA of the online network's parameters $\theta$. It outputs targets $f_\phi(o_{t+k}) \in \mathbb{R}^N$ that are used to train the online network. After each training step, the target network's weights are updated via an EMA update $\phi \leftarrow \alpha\phi + (1-\alpha)\theta$ where $\alpha$ is the target network EMA parameter. A sketch of the neural architecture is provided in Fig. 1, with more details in App. A.

**(iii) Online Network Loss Function.** Suppose our RL agent collected a batch of trajectories $\left((o_t^j, a_t^j)_{t=0}^{T-1}\right)_{j=0}^{B-1}$, where $T \in \mathbb{N}^*$ is the trajectory length and $B \in \mathbb{N}^*$ is the batch size. Then, the loss $\mathcal{L}_{\texttt{BYOL-Explore}}(\theta)$ to minimize is defined as the average cosine distance between the open-loop future predictions $g_\theta(b_{t,k}^j)$ and their respective targets $f_\phi(o_{t+k}^j)$ at time $t+k$:

$$\mathcal{L}_{\texttt{BYOL-Explore}}(\theta, j, t, k) = \left\| \frac{g_\theta(b_{t,k}^j)}{\|g_\theta(b_{t,k}^j)\|_2} - \texttt{sg}\left( \frac{f_\phi(o_{t+k}^j)}{\|f_\phi(o_{t+k}^j)\|_2} \right) \right\|_2^2,$$

$$\mathcal{L}_{\texttt{BYOL-Explore}}(\theta) = \frac{1}{B(T-1)} \sum_{j=0}^{B-1} \sum_{t=0}^{T-2} \frac{1}{K(t)} \sum_{k=1}^{K(t)} \mathcal{L}_{\texttt{BYOL-Explore}}(\theta, j, t, k),$$

where $K(t) = \min(K, T-1-t)$ is the valid open-loop horizon for a trajectory of length $T$ and $\texttt{sg}$ is the stop-gradient operator.

**(iv) World Model Uncertainties** The uncertainty associated to the transition $(o_t^j, a_t^j, o_{t+1}^j)$ is the sum of the corresponding prediction losses:

$$\ell_t^j = \sum_{p+q=t+1} \mathcal{L}_{\texttt{BYOL-Explore}}(\theta, j, p, q),$$

where $0 \le p \le T-2$, $1 \le q \le K$ and $0 \le t \le T-2$. This accumulates all the losses corresponding to the world-model uncertainties relative to the observation $o_{t+1}^j$. Thus, a timestep receives intrinsic reward based on how difficult its observation was to predict from past partial histories.

**Intuition on why $\texttt{BYOL-Explore}$ learns a meaningful representation.** The intuition behind $\texttt{BYOL-Explore}$ is similar in spirit to the one behind $\texttt{BYOL}$. In early training, the target network is initialized randomly, and so $\texttt{BYOL-Explore}$'s online network and the closed-loop RNN are trained to predict random features of the future. This encourages the online observation representation to capture information that is useful to predict the future. This information is then distilled into the target observation encoder network through the EMA slow copy mechanism. In turn, these features become targets for the online network and predicting them can further improve the quality of the online representation. For further theoretical and empirical insights on why the bootstrap latent methods learn non-trivial representations see, e.g., [72, 76].

### 3.3 Reward Normalization and Prioritization Scheme

**Reward Normalization.** We use the world model uncertainties $\ell_t^j$ to generate an intrinsic reward. To counter the non-stationarity of the uncertainties during training, we adopt the same reward normalization scheme as $\texttt{RND}$ [8] and divide the raw rewards $((\ell_t^j)_{t=0}^{T-2})_{j=0}^{B-1}$ by an EMA estimate of their standard deviation $\sigma_r$. The normalized rewards are $\ell_t^j / \sigma_r$. Details are provided in App. A.3.

**Reward Prioritization.** In addition to normalizing the rewards, we can optionally prioritize them by optimizing only the rewards with highest uncertainties and nullifying rewards with the lowest uncertainties. Because of the transient nature of the intrinsic rewards, this allows the agent to focus first on parts of the environment where the model is not accurate. Later on, if the previously nullified rewards remain, they will naturally become the ones with highest uncertainties and be optimized. This mechanism allows the agent to optimize only the source of high uncertainties and not optimize all sources of uncertainties at once. To do so, let us denote by $\mu_{\ell/\sigma_r}$ the adjusted EMA mean relative to the successive batch of normalized rewards $((\ell_t^j/\sigma_r)_{t=0}^{T-2})_{j=0}^{B-1}$. We use $\mu_{\ell/\sigma_r}$ as a clipping threshold separating high and low-uncertainty rewards. Then, the clipped and normalized reward that plays the role of intrinsic reward is: $r_{i,t}^j = \max(\ell_t^j/\sigma_r - \mu_{\ell/\sigma_r}, 0)$.

### 3.4 Generic RL Algorithm and Representation Sharing

`BYOL-Explore` can be used in conjunction with any RL algorithm for training the policy. In addition to providing an intrinsic reward, `BYOL-Explore` can further be used to shape the representation learnt by the RL agent by directly sharing some components of the `BYOL-Explore` world model with the RL model. For instance, consider a recurrent agent composed of an encoder $f_\psi$, an RNN cell $h_\psi^c$, a policy head $\pi_\psi$ and a value head $v_\psi$ that are shaped by an RL loss. Then, we can share the weights $\theta$ of the `BYOL-Explore` world model and the weights $\psi$ of the RL model at the level of the encoder and the RNN cell: $f_\psi = f_\theta$ and $h_\theta^c = h_\psi^c$ and let the joint representation be trained via both the RL loss and `BYOL-Explore`. In our experiments, we will show results for both the shared and unshared settings. Architectural details are provided in Appendix A.

## 4 Experiments

We evaluate the algorithms on benchmark task-suites known to contain hard exploration challenges. These benchmarks have different properties in terms of the complexity of the observations, partial observability, and procedural generation, allowing us to test the generality of our approach.

**Atari Learning Environment [6].** This is a widely used RL benchmark, comprising approximately $50$ `Atari` games. These are 2-D, fully-observable, (fairly) deterministic environments for most of the games but have a very long optimization horizon (episodes last for an average of $10000$ steps) and complex observations (preprocessed greyscale images which are $84 \times 84$ byte arrays). We select the $10$ hardest exploration games [5] to conduct our experiments: `Alien`, `Freeway`, `Gravitar`, `Hero`, `Montezuma's Revenge`, `Pitfall`, `Private Eye`, `Qbert`, `Solaris` and `Venture`.

**Hard-Eight Suite [22].** This benchmark comprises 8 hard exploration tasks, originally built to emphasize the difficulties encountered by an RL agent when learning from sparse rewards in a procedurally-generated 3-D world with partial observability, continuous control, and highly variable initial conditions. Each task requires the agent to interact with specific objects in its environment in order to reach a large apple that provides reward (see Fig. 2). Being procedurally-generated, properties such as object shapes, colors, and positions are different every episode. We provide videos in the supplementary materials to ground the difficulty of these tasks. Note that the current best RL agents that solve these tasks require a small (but non-zero) amount of human expert demonstrations. Without demonstrations or reward shaping, state-of-the-art deep RL algorithms, such as R2D2 [38], do not get positive reward signal on any of the tasks. In our case, we train a single RL agent and a single world model to tackle the 8 tasks all-together, making for a challenging multi-task setting.

### 4.1 Experimental Setup

At a high level, `BYOL-Explore` has 4 main hyper-parameters: the target network EMA parameter $\alpha$, the open-loop horizon $K$, choosing to clip rewards and to share the `BYOL-Explore` representation with the RL network. To better understand what part of `BYOL-Explore` is essential to perform well, we run 4 ablations. Each ablation corresponds to `BYOL-Explore` where only one hyper-parameter has been changed. The 4 ablations are namely *Fixed-targets* where the target network EMA parameter is set to $\alpha = 1$, *Horizon=1* where the horizon is set to $K = 1$, *No clipping* where we do not use clipping for the intrinsic rewards and *No sharing* where we trained separately the RL network and the

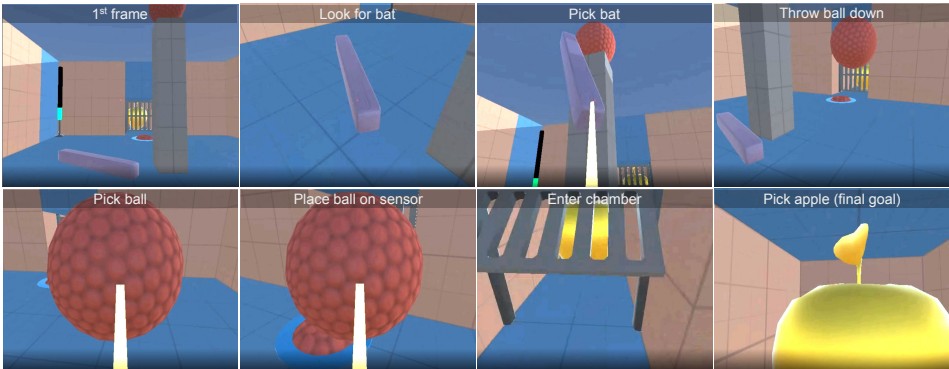

Figure 2: 1st-person-view snapshots of the human player solving `Baseball` task. They are ordered chronologically from left to right and top to bottom. Each image depicts a specific stage of the task.

`BYOL-Explore`'s world model. In addition to `BYOL-Explore`, we also run as prominent baselines `RND`, `ICM` (see App. B for details), and pure RL which is an RL agent only using extrinsic rewards.

Finally, we run experiments on two different evaluation regimes. The first regime uses a mixed reward function $r_t = r_{e,t} + \lambda r_{i,t}$ which is a linear combination of the normalized extrinsic rewards $r_{e,t}$ and intrinsic rewards computed by the agent $r_{i,t}$ with mixing parameter $\lambda$. This may be the most important regime for a practitioner as we can see if our intrinsic rewards help improve performance, with respect to the extrinsic rewards, compared to the pure RL agent. The second regime is fully self-supervised where only the intrinsic reward $r_{i,t}$ is optimized. This regime gives us a sense of how pure exploration methods perform in complex environments.

**Choice of RL algorithm.**  We use VMPO [64] as our RL algorithm. VMPO is an efficient on-policy optimization method that has achieved strong results across both discrete and continuous control tasks, and is thus applicable to all of the domains we consider. Further details regarding the RL algorithm setup and hyperparameters are provided in Appendix C.

**Performance Metrics.**  We evaluate performance in terms of the agent score at a number of observations/frames $t$, $\texttt{Agent}_{\text{score}}(t)$, as measured by undiscounted episode return. The number of frames $t$ corresponds to all the frames generated by all the actors by interacting with the environment, even the skipped ones. Frames/observations can be skipped if there is an action repeat which is the case in `Atari` where the action repeat is of $4$.

We define the highest agent score through training as $\texttt{Agent}_{\text{score}} = \max_t \texttt{Agent}_{\text{score}}(t)$, as done in [18, 3]. We define, for each game, the Human Normalized Score (HNS) at number of frame $t$: $\texttt{HNS}(t) = \frac{\texttt{Agent}_{\text{score}}(t) - \text{Random}_{\text{score}}}{\text{Human}_{\text{score}} - \text{Random}_{\text{score}}}$ as well as the HNS over the whole training: $\texttt{HNS} = \max_t \texttt{HNS}(t)$. A HNS higher than 1 means superhuman performance on a specific task. We similarly define the `CHNS` Score as HNS clipped between 0 and 1.

### 4.2   Atari Results

In these experiments, we set the target EMA rate $\alpha = 0.99$ and open-loop horizon $K = 8$. We use $\lambda = 0.1$ to combine the intrinsic and extrinsic rewards. We follow the classical 30 random no-ops evaluation regime [46, 73], and average performance over 10 episodes and over 3 seeds. This evaluation regime does not use sticky actions [43].

Fig. 3 (left) shows that `BYOL-Explore` is almost superhuman on the 10-hardest exploration games and outperforms the different baselines of `RND`, `ICM`, and pure RL. Fig. 3 (right) compares `BYOL-Explore` against its ablations to gain finer insights into our method. The *No clipping* ablation performs comparably, showing that the prioritization of intrinsic rewards is not necessary on `Atari` tasks. Similarly, the *Horizon=1* ablation performs slightly better, indicating that simply predicting one-step latents is sufficient to explore efficiently on the fully-observable `Atari` tasks. The *Fixed Targets* ablation performs much worse, showing that our approach of predicting learned targets (rather than

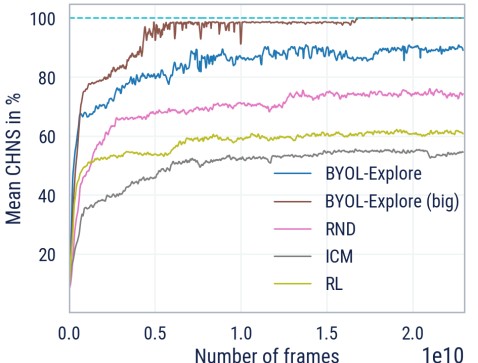 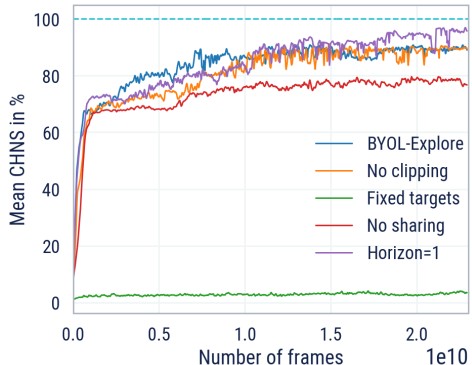

Figure 3: Mean CHNS($t$) score across the tasks in `Atari`. **Left**: `BYOL-Explore` and the baselines in the mixed regime for `Atari`. **Right**: `BYOL-Explore` and its ablations in the mixed regime.

fixed random projections) is vital for good performance. It is also worth noting that all the ablations except *Fixed Targets* outperform all of our baselines, demonstrating the robustness of our approach.

Finally, because the *Horizon=1* ablation was close to superhuman on `Atari`, we run the same configuration but double the length of the sequences on which we train from 64 to 128 (also doubling memory requirements while learning). With this small adjustment, this agent (`BYOL-Explore` (big)) becomes superhuman on all of the 10-hardest exploration games.

**Purely intrinsic exploration.** We test how `BYOL-Explore` behaves when only given intrinsic rewards without any extrinsic signal on the well-known `Montezuma's Revenge` game. We measure exploratory behavior in terms of the number of different rooms of the dungeon the agent is able to explore over its lifetime. Note that accessing later rooms requires navigating complex dynamics such as collecting keys to open doors, avoiding enemies, and carefully traversing rooms filled with traps such as timed lasers. Figure 4 shows how much room coverage is achieved during training when no extrinsic reward is used, showing that `BYOL-Explore` explores further than the best result reported by `RND` [8]. Importantly, we use the episodic setting for intrinsic rewards whereas the published `RND` results considers the non-episodic setting for intrinsic rewards — facilitating exploration as the agent is less risk-averse. Therefore, our setting could be considered even more challenging. Our agent explores more than 20 rooms on average versus 17 with best published `RND` results. As expected in the

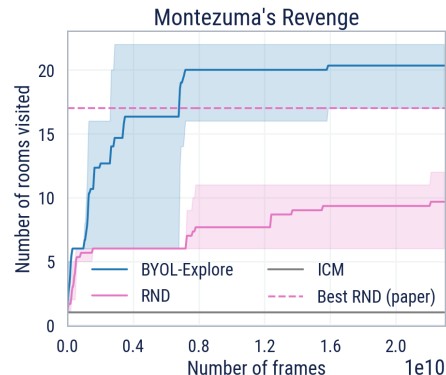

Figure 4: Number of rooms visited in `Montezuma's Revenge` during training in the self-supervised regime over 3 seeds.

episodic setting, our `RND` re-implementation visits even fewer rooms. However, we can reproduce the published `RND` results in the episodic setting when using recurrent policies.

**Further results.** More fine-grained results are reported in App D.1. We report, in Fig.11 and in Fig.12, the agent scores learning curves for each game. Tab. 1 and Tab. 2 have agent score at the end of training. Finally, Tab. 3 and Tab. 4 show the mean CHNS and different statistics (mean and percentiles) of the HNS across the selected games. An interesting finding from examining the HNS is that clipping and longer-horizon predictions are critical for very high scores on some games such as `Montezuma's Revenge` or `Hero`. `BYOL-Explore` has a median HNS of 331.98 compared to the *No-clipping* ablation and the *Horizon=1* which have a median HNS of only 181.39 and 199.80 respectively. Therefore, while clipping is not necessary to get to human-level performance, it is still crucial to achieve top performance. We also provide further results regarding the pure exploration setting on all 10 games in App. D.2.

### 4.3 Atari's results in the Presence of Stochastic Distractors

Stochastic distractors or noise are well known to break intrinsic curiosity methods based on prediction error of future frames. This is because the prediction squared-error loss of a future frame has an irreducible component which corresponds to the variance of the future frame distribution [51, 52]. However, because BYOL-Explore is not a prediction error method at the frame-level but at the latent level, we can hope that some noise present in the frame can be removed from the latent embedding. More specifically, we hypothesize that BYOL-Explore removes noisy features of the frame that are not useful to minimize the BYOL-Explore loss in order to better minimize this loss. Those are features that are not useful for future predictions. On the other hand, since RND uses a random network to build it targets, RND by construction cannot actively remove noisy features from its targets. To better show this, we slightly changed the Atari environment to generate noisy frames. More precisely, the new observation is made of two parts of the same size ($84 \times 84$), on the right side we have noise and on the left we have the original Atari image. Each time the no-op action is chosen by the agent, a new noise is sampled. The noise is composed of $14 \times 14$ blocks of $6 \times 6$ pixels. Each block of pixel is assigned a uniform random value between $0$ and $255$ if a new noise is sampled. We provide a frame of this modified Atari environment in Fig. 5 (left).

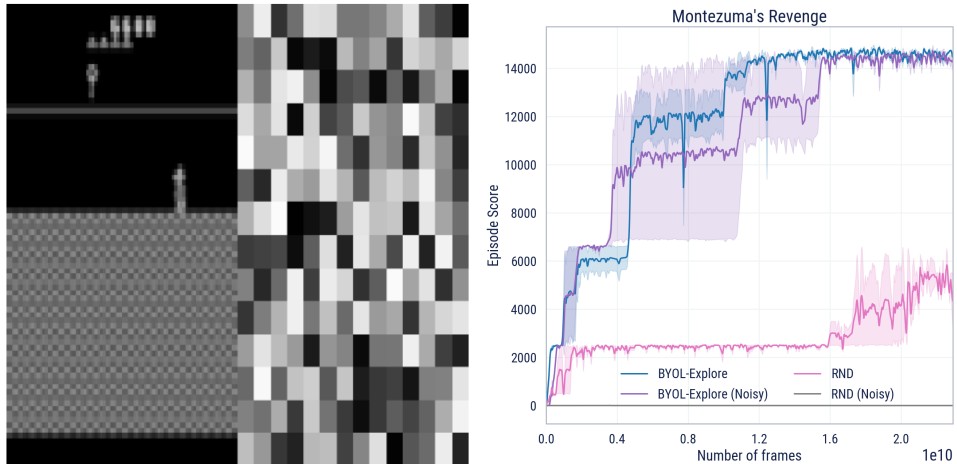

Figure 5: Experiments on noisy Atari. **Left**: Noisy Atari environment. **Right**: Learning curves in terms of agent score for BYOL-Explore and RND for noisy and normal Atari on Montezuma's Revenge, averaged over 10 episodes and 3 seeds.

We train BYOL-Explore and RND on this noisy and normal version of Atari on the game Montezuma's revenge (we do not include a comparison with ICM because it was already not performing well in the normal version). To get better results for BYOL-Explore, we increase the predictor to have 3 hidden layers of $512$ instead of $1$ hidden layer of $256$. The results are reported in Fig. 5 (right). We observe that BYOL-Explore is perfectly able to deal with that type of controllable-noise whereas RND is not and completely flat-lined in the noisy environment because the agent is attracted to the noise and keeps repeating the no-op action.

### 4.4 DM-HARD-8 Results

In these experiments, we set the target EMA rate $\alpha = 0.99$ and open-loop horizon $K = 10$. We use $\lambda = 0.01$ to combine the intrinsic and extrinsic rewards. In contrast to prior work [22], we perform experiments in the more challenging multi-task regime, training a single agent to solve all eight tasks. At the beginning of each episode, a task is drawn uniformly at random from the suite.

In Fig. 6 (left) we report the mean CHNS($t$) across the tasks, averaged over 3 seeds. We see that BYOL-Explore outperforms the baselines of RND, ICM, and pure RL by a large margin. Fig. 6 (right) compares the performance of BYOL-Explore to its various ablations. Note that the *No-clipping* ablation performs similarly to BYOL-Explore in terms of CHNS. However, unlike the fully-observable Atari tasks, the *Horizon=1* ablation learns considerably slower and achieves lower final performance (see also our extended ablations on the horizon length in Fig. 15 in App. D.3). We note once again that

the `BYOL-Explore` bootstrapping mechanism for learning representations is essential, as confirmed by the poor performance of the *Fixed-targets* ablation. Due to computational limitations, we did not run the *No Sharing* ablation, as using separate networks requires twice the memory.

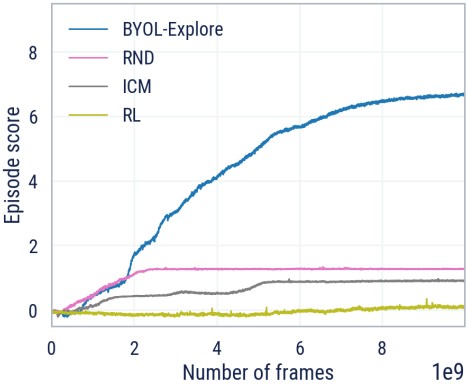 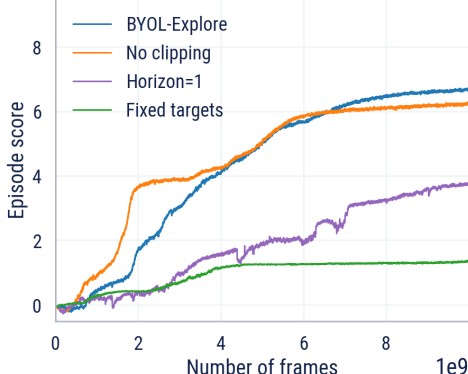

Figure 6: Mean $\text{CHNS}(t)$ score across the tasks in the `DM-HARD-8` suite. **Left**: `BYOL-Explore` against baselines: `ICM`, `RND` and Pure RL. **Right**: `BYOL-Explore` against various ablations.

We now analyze our method more closely by examining per-task performance. The full learning curves for each task can be found in Fig. 7 for `BYOL-Explore` and the main baselines and in Appendix D.3 (see Fig. 14) for the various ablations. First, we take note that other curiosity-driven methods (`ICM` and `RND`) cannot get any positive score on the majority of the `DM-HARD-8` tasks, even with additional hyperparameter tuning and reward prioritizing (see Fig. 17 and Fig. 18 in App. D.3).

In contrast, we see that `BYOL-Explore` achieves strong performance on five out of the eight hard exploration tasks. Importantly, `BYOL-Explore` achieves this without human demonstrations, which was not the case in prior work [22]. `BYOL-Explore` even surpasses humans on 4 tasks, namely `Navigate cubes`, `Throw-across`, `Baseball`, and `Wall Sensors` (see Tab. 9 in App. D.3 for details). Most impressively, `BYOL-Explore` can solve `Throw-across`, which is a challenging task even for a skilful human player and was not solvable in prior work without collecting additional successful human demonstrations [22].

Interestingly, note that on the `Navigate Cubes` task, both `RND` and the *Fixed-targets* ablation achieve maximum performance alongside `BYOL-Explore`. We argue that this is because the prediction of random projections (either at the same step as done by `RND` or multi-step as done by `BYOL-Explore`) leads to the policy learned performing spatial, navigational exploration — this is the kind of behavior required to explore well on the `Navigate Cubes` task. In contrast, the other tasks require exploratory behavior involving interaction with objects and the use of tools, where both `RND` and the *Fixed-targets* ablation fail. Finally, we observe that two games, namely `Remember Sensor` and `Push Blocks`, are particularly challenging, where all of our considered methods perform poorly. We hypothesize that this is due to the larger variety of procedurally generated objects spawned in these levels, and the need to remember previous cues in the environment leading to a hard credit assignment problem.

**Purely intrinsic exploration.** Each of the `DM-HARD-8` tasks has complex dynamics and object interactions, making it difficult to assess qualitatively the behavior of purely intrinsically motivated exploration. Nevertheless, for completeness, we provide results of `BYOL-Explore` trained only with intrinsic rewards in App. D.3, showing that it does achieve some positive signal on the `Drawbridge` and `Wall Sensor` tasks (see Fig. 19).

## 5 Conclusion

We showed that `BYOL-Explore` is a simple curiosity-driven exploration method that achieves excellent performance on hard exploration tasks with fairly deterministic dynamics. `BYOL-Explore` is a multi-step prediction error method at the latent level that relies on recent advances in self-supervised learning to train its representation as well as its world-model without any additional loss. In `Atari`, `BYOL-Explore` achieves superhuman performance on the 10-hardest exploration games while being

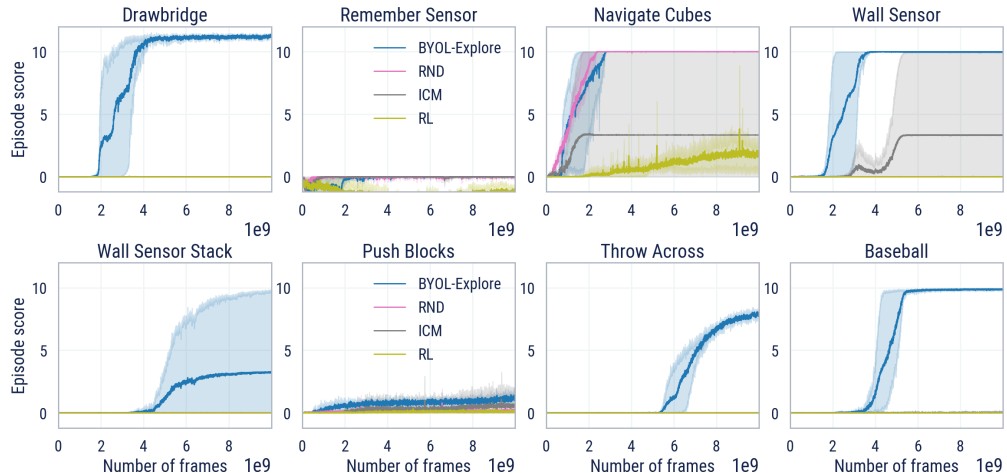

Figure 7: Agent's score for each task in the `DM-HARD-8` suite for `BYOL-Explore` against baselines. Shaded areas correspond to the minimum and maximum values across three seeds.

of much simpler design than other superhuman agents. Moreover, `BYOL-Explore` substantially outperforms previous exploration methods on `DM-HARD-8` navigation and manipulation tasks in a 3-D, multi-task, partially-observable and procedurally-generated environment. This shows the generality of our algorithm to handle either 2-D or 3-D, single or multi-task, fully or partially-observable environments. While one main limitation of our algorithm is that it is not designed to fully handle stochasticity in environments (e.g. sticky actions), we show that `BYOL-Explore` is robust to some simple kinds of controllable noise ('TV-noise') since operating in latent space allows the latent representation to filter it out.

In the future, we would like to improve performance in `DM-HARD-8` and to demonstrate the generality of our method by extending it to other domains. In `DM-HARD-8`, we believe we can improve performance by scaling up the world model and finding better ways to trade off exploration and exploitation. Beyond `DM-HARD-8`, there are opportunities to tackle further challenges, most notably highly-stochastic and procedurally-generated environment dynamics such as `NetHack` [40]. To do so, we are investigating different mechanisms to adapt prediction-based methods to stochastic environments in order to only use the epistemic uncertainty as an intrinsic reward and discard the aleatoric uncertainty.

## Acknowledgments and Disclosure of Funding

We would like to thank Abbas Abdolmaleki, Arunkumar Byravan, Adrià Puidomenech Badia, Tim Harley, Steven Kapturowski, Thomas Keck, Jean-Baptiste Lespiau, Kat McKinney, Kyriacos Nikiforou, Georg Ostrovski, Razvan Pascanu, Doina Precup, Satinder Singh, Hubert Soyer, Pablo Sprechmann, and Karl Tuyls for their support and advice in developing and publishing this work.

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
