# OpenReview forum: "BYOL-Explore: Exploration by Bootstrapped Prediction"
_NeurIPS.cc/2022/Conference — NeurIPS 2022 Accept_

### Official Review · Reviewer_SjCF · 2022-07-06

**Rating:** 7
**Confidence:** 4
**Soundness:** 3 good
**Presentation:** 3 good
**Contribution:** 3 good

**Summary:**

The paper presents a new way of providing intrinsic rewards for exploration, by learning a latent space and predicting its future latents. The intrinsic reward is then proportional to the error between the predicted and actual future latents, so that states that are harder for the world model to predict are rewarded more. The method achieves impressive results on two hard-exploration benchmarks: hard-exploration Atari games and DM-HARD-8.

**Questions:**

See above, reiterating here:

- What is your sample complexity?
- What form of stochasticity did you use on Atari (sticky actions, no-ops, nothing…)?
- Why do you think your RND performance appears lower than the RND paper?

**Limitations:**

One limitation that I would have liked discussed is the paper’s dependence on determinism or lack thereof. The RND paper tries to solve the following “noisy-TV problem” with sticky actions: suppose you are in room 1 of Montezuma’s and room 2 is immediately to your right, then depending on how the randomness of sticky actions work out, it may be that you only have a 50% chance of getting to room 2 during the next frame, and a 50% chance of staying in room 1. As a result, if you try to predict some feature of *future* frames for your exploration bonus, the exploration bonus may stay high for those stochastic states no matter how often they are explored, which is why RND tries to predict features of the *current* frame instead. Given that this work predicts features of future frames, it may be vulnerable to the noisy-TV problem stated above. The authors should either disclose this as a limitation or explain why they think their work  avoids this issue.

**Strengths And Weaknesses:**

 Strengths:

The work shows impressive performance on Atari, and seemingly on DM-HARD-8 (note: I am very familiar with past Atari results so I can judge those at a glance. For DM-HARD-8 I am relying on the comparison included in the paper, hence the more qualified statement). I am particularly impressed by the performance on Pitfall, given in the appendix (IMO the authors should call that out more, as Pitfall is a harder game than Montezuma’s, and their performance improvement relative to, say, RND is larger on Pitfall than Montezuma’s).

I agree with the authors’ claim that the method is simpler than Go-Explore and Agent57, so it seems to represent a genuine improvement for the field.

Weaknesses:

The main weakness is that the paper fails to make standard declarations about how they ran the benchmarks, at least in the case of Atari (which again I am most familiar with). While I do think the work would be at least somewhat impressive no matter what these specifics happen to be, without them it is really hard to decide if the work is an incremental improvement or a game-changer in the field.

Specifically:

- The number of sample frames is never stated. Instead, the x-axis of all plots is given in “Learner Steps” which is, frankly, a meaningless number. All Atari work I am aware of (including works that this work compares to like RND, Agent57, and Go-Explore) states the number of frames used in some form (note: it is standard practice to include skipped frames if using frame skipping, when updating, the authors should make sure they clarify their frame skipping strategy and whether it is included in their frame count). Without this number, it is hard to compare the method to, say, Agent57. Further, Figure 4 seems to indicate that the authors’ implementation of RND might perform substantially worse than that in the RND paper, but it is hard to tell if this is due to differences in sample complexity. If it is not, it would be interesting for the authors to briefly discuss why they think their RND implementation performs worse.
- Likewise, it seems to have been standard prior to this work to give the sample complexity on DM-HARD-8 as well, so the authors should also report it.
- No information as far as I can tell is given about how stochasticity is injected in the Atari results (no-ops, sticky actions, etc.). This has been a contentious point historically, with some evidence from Machado et al. [1] that sticky actions reduce average performance on Atari. Without knowing if they were used, it is hard to properly place the results. Further, RND (which is a primary comparison of this paper) was specifically designed to solve noisy-TV issues that arise due to sticky actions, so it would be useful to understand if this method is vulnerable to those issues (which I think it might be, see Limitations section).

I recommend the authors consult Machado et al. [1] for a set of evaluation recommendations for Atari. While I personally don’t expect every Atari paper to follow this exact evaluation protocol (the recommended frame complexity is too low IMO, and while I believe sticky actions etc. should generally be followed, I am conscious of the fact that they aren’t universally adopted), I do think deviations from these recommendations are important to note explicitly, and they usually are in Atari work I am familiar with.

Overall, my main request for this paper is for the frame complexity and the evaluation method for Atari (especially the way stochasticity is injected) to be clarified. The method is interesting enough and the performance is good enough that I will switch my rating to somewhere within the “accept” range when this information is provided (where exactly would depend in part on what this information is specifically: if the method uses, say, 10x fewer frames than Agent57, then that is a lot more impressive than if it uses around the same amount). However, I think it would do a disservice to the community to accept the paper in the current state, since these issues make it very hard to compare the results to competing methods.

[1] Machado, Marlos C., et al. "Revisiting the arcade learning environment: Evaluation protocols and open problems for general agents." 2017. *Journal of Artificial Intelligence Research*



UPDATE: As promised, I am switching my rating to an "Accept" given that the information I requested was provided. My choice of going to "Accept" and not higher is primarily due to the information being in line with what I expected (frame complexity similar to Agent57, no sticky actions), which still qualifies these results as very solid and worthy of NeurIPS, but not absolutely groundbreaking.

---

> ### Author Response · Authors · 2022-08-01
> **Author Feedback**
>
> We thank the reviewer for their comprehensive review. The reviewer has comments regarding:
>  - Sample complexity
>  - Form of stochasticity
>  - Performance of our RND implementation
>  - Noise, stochastic distractors
>
> Concerning sample complexity, (1) we decided to use learner steps because this is what used in the baselines papers we  compare to, namely RND (https://arxiv.org/pdf/1810.12894.pdf) and ICM (https://pathak22.github.io/noreward-rl/resources/icml17.pdf).  However, we think that the reviewer's suggestion of using the number of frames instead of learner steps is pertinent and we have updated those curves in the rebuttal version to allow easy comparison to other agents. For the number of frames, we count every frame, even the skipped ones (e.g., in Atari we have an action repeat of 4 which means that we skip 3 frames every 4).
>
> (2) our data efficiency is overall on par with Agent57. We did not investigate improving sample complexity beyond this point, as sample complexity can depend greatly on a variety of agent implementation details such as neural architectures, stochastic training, and reusing collected data multiple times [1] which are orthogonal to our contribution.
> [1]: MuZero Reanalyze https://arxiv.org/pdf/1911.08265.pdf
>
> Concerning the form of stochasticity used in our evaluation, we explicitly say line 214: “We follow the classical 30 random no-ops evaluation regime [44, 70]”. We cite papers following the same protocol and where the reader can find more details. As this evaluation regime is described in detail in several papers DQN, R2D2,.. we skip it here. In this evaluation regime we do not use sticky actions, we will make this clear in the rebuttal draft. Actually sticky actions introduces a stochasticity that our exploration method is not built for (as explained in Reviewer Yjyb’s response) and we will address this issue in future work. For now, our algorithm is fundamentally based on prediction-error at the latent level and therefore can be subject to stochastic traps. However, BYOL-Explore is still able to get rid of noise that does not fundamentally impact the dynamics as shown in Appx. A.4.3. Finally, here our main focus is on exploration in deterministic environments and we consider extending our work to the environment with stochastic dynamics as a future work (as mentioned in the conclusion section line 308) .
>
> Concerning the performance of our RND implementation, we have a discussion on this particular point in line 244-251. In the RND paper, the best results are reported with a non-episodic setting for the intrinsic reward, a double value head and a non-recurrent policy (called CNN policy in the RND paper). As our RL agent uses the normal episodic setting, a single value head and recurrent policies and that we just interchange the exploration algorithm between BYOL-Explore, ICM and RND for fair comparison of the algorithms, then our RND implementation corresponds to the episodic, with single-value head with RNN policy of the RND paper (see Fig 4(a) of the original RND paper https://arxiv.org/pdf/1810.12894.pdf). This particular instance of RND gets a little bit more than 4K in the original paper which is exactly the results that we get with our own RND implementation. This makes us quite confident that we reproduce this baseline correctly.
>
> Concerning stochastic distractors which was also a comment of Reviewer Yjyb, here was our response: Stochastic distractors can be of different types ranging from white/controllable noise added to the observations to distractors that completely change the underlying dynamics of the environment such as sticky actions. BYOL-Explore will be able to handle the former but not the latter. Indeed, at its core, BYOL-Explore  is a prediction-error method at the latent level. Therefore, it will be subject to stochastic traps if the possible future latents are not deterministic, which will happen with sticky actions since different observations can be generated from a given past and an action.
>
> However, if the noise is not changing fundamentally the dynamics and the future possible observations are just noisy versions of each other then BYOL-Explore will be able to filter the noise because this is not a useful feature that allows to predict efficiently the future frames.
> To experimentally show this property of BYOL-Explore, we have run some experiments with controllable noise added to the observations. More precisely, the new observation is made of two parts of the same size (84x84), on the right side we have noise and on the left we have the original Atari image. Each time the no-op action is chosen by the agent, a new noise is sampled. The noise is composed of 14x14 blocks of 6x6 pixels. Each block of pixel is assigned a uniform random value between 0 and 255 if a new noise is sampled.
> We observe that BYOL-Explore is perfectly able to deal with that type of controllable noise whereas e.g., RND is not. These results are summarized in Appendix A.4.3.

---

> > ### Comment · Reviewer_SjCF · 2022-08-04
> > **Thanks for the clarifications**
> >
> > I thank the authors for their clarifications, and my apologies to the extent that I missed some of the things that I asked about in the first place. I have switched my rating to an "Accept".
> >
> > Concerning the frame count, I would note that although the RND paper does not put the frame count on the x axis, it does state it several times in the text, which the original version of this paper did not do. I agree that the ICM paper does not give the frame count at all (even though in my opinion it should), but note that it is somewhat of an "old" paper (I can't believe I am saying this of a paper from 2017, but it is kind of true in our field) and most importantly is tested on environments that have a lot less historical contention.
> >
> > Concerning the type of stochasticity and the difficulty of handling sticky actions, while I think it is an important limitation of the work, I also don't feel like it should be a dealbreaker, as assuming roughly deterministic dynamics is a reasonable assumption in many (but not all!) cases in my opinion. I hope the authors can figure out how to handle such noise in the dynamics in the future, though I think this may be difficult. Regardless, this shouldn't block acceptance of the current version this year IMO.

---

### Official Review · Reviewer_YjyB · 2022-07-09

**Rating:** 7
**Confidence:** 3
**Soundness:** 3 good
**Presentation:** 4 excellent
**Contribution:** 2 fair

**Summary:**

The authors present a Bootstrap-your-own-latent inspired method for model learning to generate intrinsic rewards for exploration. The method consists of 2 parallel networks, one an exponential moving average of the other with goal of predicting k steps into the future conditioned on past actions, current state and k future actions. The predictions occur on the latent space with a cosine loss for training. The discrepancy between the predictions is used to generate rewards for the RL agent to induce exploration. The authors test their method on DM-Hard and hard exploration games in Atari and show considerable performance improvement with reference to curiosity baselines.


**Questions:**

How does it work in presence of distractors? Some experiments of the "noisy TV" would be nice to see.


**Limitations:**

The paper is principled and along the lines of previous works. The results are impressive and useful for the community. The evaluation is thorough in my opinion.

**Strengths And Weaknesses:**

Strengths:
1. Method is principled. The intuition behind why it works is clear from previous works.
2. Well-written and easy to follow: The authors describe the method well along with the setup.
3. Extensive experimentation: The experimentation suite is large, with experiments in first-person ego-centric environments as well as hard exploration problems in third-person atari games.

Weaknesses:
1. Not a significant algorithmic development: The method proposed is not necessarily. While this is not a weakness on its own, it can be said that the contribution is not algorithmic development.
2. Missing some related works: http://proceedings.mlr.press/v139/sontakke21a/sontakke21a.pdf and  https://arxiv.org/pdf/2109.08603.pdf among many many others. Please do a more thorough search and add relevant citations.

---

> ### Author Response · Authors · 2022-08-01
> **Author feedback**
>
> We thank the reviewer for their comments, questions and suggestions.
>
> The main question was about stochastic distractors in the environment. Stochastic distractors can be of different types ranging from white noise added to the observations, to distractors that completely change the underlying dynamics of the environment, such as sticky actions in Atari. BYOL-Explore will be able to handle the former but not the latter.
> Indeed, at its core, BYOL-Explore  is a prediction-error method at the latent level. Therefore, it will be subject to stochastic traps if the possible future latents are not deterministic, which will happen e.g. with sticky actions since completely different observations can be generated from a given history and an action.
>
> However, if the noise is not changing fundamentally the dynamics and the future possible observations are just noisy versions of each other then BYOL-Explore will be able to filter the noise because this is not a useful feature that allows to predict efficiently the future frames.
> To experimentally show this property of BYOL-Explore, we have run some experiments with controllable noise added to the observations of Montezuma Revenge.  More precisely, the new observation is made of two parts of the same size (84x84), on the right side we have noise and on the left we have the original Atari image. Each time the no-op action is chosen by the agent, a new noise is sampled. The noise is composed of 14x14 blocks of 6x6 pixels. Each block of pixel is assigned a uniform random value between 0 and 255 if a new noise is sampled. We observe that BYOL-Explore is perfectly able to deal with that type of stochastic distractor (controllable-noise)  and achieve the same performance as the deterministic Montezuma whereas, for instance, RND completely fails. These results are summarized in Appendix A.4.3.
>
> Finally, we have added the missed citations in the rebuttal version.

---

### Official Review · Reviewer_huAP · 2022-07-10

**Rating:** 7
**Confidence:** 3
**Soundness:** 4 excellent
**Presentation:** 4 excellent
**Contribution:** 3 good

**Summary:**

This paper builds on the BYOL work from 2020 and explores using it in the context of curiosity-driven exploration. It learns a world model via an RNN + self-supervised loss mechanism. This mechanism, optionally with an external task loss, is used to train an RL policy in order to play Atari and DM-Hard-8. Besides presenting superhuman results on the ten hardest Atari games, they also present favorable exploration results on Montezuma's Revenge. Notably, they do not use any human demonstrations.

They compare to other works in the curiosity-driven space on the metric of mean CHNS (clipped human normalized score). Across all ablations except the one fixing the EMA update to not update, their reported method outperforms the baselines and is superhuman when allowed to train on longer sequences.

They show the exploration results on Montezuma's Revenge to test what happens in the purely intrinsic exploration setting. Because there are no external reward results given, we should not expect the model to beat the game but instead to explore the setting. Compared to the curiosity-driven baselines, it does that, finding 20 rooms compared against 17 in the published results.

**Questions:**

I don't have any major questions other than the one about the lambda sensitivity. It would be cool if you could keep going on the big model to see how it does in DM-Hard-8 as well, but I'd understand if the time/compute budget isn't there.

**Limitations:**

Yes, they understand that their model is quite a ways off from being realistically used yet. They talk in the conclusion about improving on DM-Hard-8 and then moving on to more difficult environments like NetHack.

**Strengths And Weaknesses:**

This paper is good. There doesn't appear to be anything world-changing in it, but it's a solid paper presenting a good idea and written well.

Originality:

The idea of forward prediction is not new, nether is BYOL, the surprisingly non-degenerative self-supervised model on which BYOL-Explore is built. Combining these concepts by using an RNN to do the prediction makes sense. This isn't original and I'm glad someone tried it.

Quality:

The quality of this paper is high. I don't see any experiments that I need to have run, nor do I see any major faults with their reasoning. This is a good piece of work. That being said, I certainly would _like_ to see some other experiments for my own interest :).

An example is what happens as you scale up or down the lambda on the DM-8 experiments. It's quite low at 0.01 right now and so I'm left wondering if that's just for optimization reasons or if the model actually just doesn't like to solve tasks as that gets larger. If it was an order of magnitude lower, would it succeed a lot better? In other words, what's the sensitivity of this parameter as it's a crucial cog in the presented algorithm.

Clarity:

This paper is clear and well written. Thank you for that.

Significance:

The results aren't game changing, but they are of great interest to the self-supervised learning and curiosity-driven learning communities. BYOL made a large splash when it debuted and this extension of BYOL is notable for the community. Others are likely to use it in their baselines and perhaps build upon it.

I'm still surprised that the BYOL models don't collapse to a single point, so more evidence of this mechanism proving fruitful in complex self-supervised methods is helpful for building my model of how this works.

---

> ### Author Response · Authors · 2022-08-01
> **Author feedback**
>
> We thank the reviewer for their comments and suggestions. Concerning the novelty, the reviewer is right. Our work consists mainly in building a multi-step prediction error at the latent level to tackle exploration in fairly deterministic environments. This is done by combining previous works such as prediction-error and BYOL. Even if the work is not world-changing, it is important to be done and known by the community since it creates a fairly simple and performant exploratory RL agent. Nevertheless we  would like to highlight our contribution that BYOL-Explore can be used as a standalone learning algorithm to train the agent (see Sec. 4.2, pure exploration) without requiring any extrinsic reward, whereas prior work such as   Guo et. al. 20,  Schwarzer et. al. 20  have used the BYOL-style architectures as an auxiliary representation learning loss in conjunction with RL. We believe this is an important and novel contribution since it shows that even in the absence of extrinsic reward BYOL-Explore can learn a meaningful representation/exploration strategy.
>
> The main comment was about the choice of the mixing parameter in the DM-Hard 8 experiment.
> Sensitivity to λ.
> Most contemporary RL methods that optimize a weighted combination of intrinsic and extrinsic rewards rely on a proper choice of these weights.
> The optimal setting of this choice is domain-dependent and task-dependent:
> if the weight of the intrinsic reward is too low, the agent won't explore enough;
> if it's too high, it risks prioritizing exploratory behavior rather than the desired behavior that maximizes extrinsic reward (exploitation).
>
> BYOL-Explore is no exception, and it benefits from a deliberate choice of λ, but it is not too sensitive to it: In Hard Eight we found that setting λ to 1e-3 and 1e-2 struck a good balance between exploration and exploitation, with 1e-2 better overall. 1e-4 was only marginally exploratory, and 1e-1 interfered with exploitation.
> So we could see its benefits when selecting λ from a reasonable set of candidates (powers of 0.1).
> As further evidence that BYOL-Explore is not too sensitive to λ, notice that our method explored effectively in multitask Hard Eight and single task Atari with task-independent choices of λ (respectively 1e-2 and 1e-1).
>
> We believe that to go further (and remove the need to tune λ) requires a systematic study of the problem of trading off exploration and exploration, and that it merits its own independent investigation in future work.
>
> References:
>
> Schwarzer et. al. 2020. Max Schwarzer, Ankesh Anand, Rishab Goel, R Devon Hjelm, Aaron Courville, and Philip Bachman. Data-efficient reinforcement learning with self-predictive representations. arXiv preprint arXiv:2007.05929, 2020.
>
> Guo et. al. 2020. Zhaohan Daniel Guo, Bernardo Avila Pires, Bilal Piot, Jean-Bastien Grill, Florent Altché, Rémi Munos, and Mohammad Gheshlaghi Azar. Bootstrap latent-predictive representations for multitask reinforcement learning. In International Conference on Machine Learning, pages 3875–3886. PMLR, 2020

---

### Official Review · Reviewer_tiha · 2022-07-10

**Rating:** 6
**Confidence:** 3
**Soundness:** 3 good
**Presentation:** 3 good
**Contribution:** 2 fair

**Summary:**

This paper introduces BYOL-Explore, a novel exploration method for RL. The idea of BYOL-Explore is to train an encoder of observations based on BYOL and further utilize training loss as the intrinsic reward for exploration.

The architecture of BYOL-Explore consists of an observation encoder, a closed-loop RNN that embeds the history of interaction, and an open-loop RNN for future prediction. In training the observation embeddings, the authors adopt the learning scheme of BYOL and construct a targe encoder by a moving average of the trained encoders. The training of the encoder thus minimizes the distance between the RNN prediction and the target encoder of observations.

Experiments suggest that BYOL-Explore outperforms the SOTA baselines RND and ICM.

**Questions:**

$\{\bf{\rm Question 1.}\}$ The proposed RNN structure seems relatively large-scaled compared with most existing exploration strategies. How does BYOL-Explore compare with ICM and RND regarding the computation budget and training efficiency?

$\{\bf{\rm Question 2.}\}$ Does the proposed intrinsic reward of BYOL-Explore capture the epistemic uncertainty of observations adequately? For instance, if a specific observation-action pair $(o, a)$ frequently occurs among the trajectories, we would hope that the intrinsic reward of visiting $(o, a)$ is small as it is of less novelty. Nevertheless, the proposed intrinsic reward sums up all the prediction errors involving the pair $(o, a)$ across the trajectories, which might be even larger than that of scarcely explored observation-action pairs.

$\{\bf{\rm Question 3.}\}$ In addition, the proposed intrinsic reward seems to have a preference over actions that are distant from the initial observations. For instance, for a given trajectory ${x_1 = (o_1, a_1), x_2 = (o_2, a_2), \ldots, x_n = (o_n, a_n)}$, the intrinsic reward of $x_1$ consists only the loss associated with the history representation $b_{1, 1}$. Nevertheless, since each closed-loop RNN is followed by the open-loop RNN cells that reconstruct the entire trajectory, the intrinsic reward of $x_4$ thus sums up three terms, namely, the loss associated with the history representations $b_{1, 3}$, $b_{2, 2}$, and $b_{3,1}$. Will such design leads to a higher intrinsic reward of $x_4$ than $x_2$? Is such a preference for ending steps desirable for exploration?


**Limitations:**

N.A.

**Strengths And Weaknesses:**

$\{\bf{\rm Strength.}\}$

-The proposed BYOL-Explore calculates the intrinsic reward and thus is adaptable to existing RL algorithms.

-The proposed curiosity is built on future multi-step predictions, thus utilizing the temporal information contained in observations. In addition, the intrinsic reward is defined by the training loss on latent representation. If the latent well captures the temporal information of observations, the proposed intrinsic reward can filter out the reconstruction loss of irrelevant observations (to the transition dynamics), which is desirable.

$\{\bf{\rm Weakness.}\}$

-The learning paradigm of BYOL-Explore falls in the common practice of fitting (coarse) transition models and then utilizing its training loss (or the disagreement among the ensemble of models) as the intrinsic reward/penalty for exploration/conservative RL. In that regard, the contribution of this paper is relatively marginal.

---

> ### Author Response · Authors · 2022-08-01
> **Author feedback**
>
> We thank the reviewer for their comments and questions.
>
> The reviewer has indicated that the contribution of the paper is relatively marginal because the method falls within a general approach of intrinsic motivation from dynamics models. We would like to highlight that even if the contribution appears marginal, the outcome of the paper is a very simple and performant agent that is on par with much more complex agents such as Agent57. We consider this as a valuable contribution for the RL community.
>
> Concerning Q1, there is indeed an additional computational cost due mainly to unrolling the openloop RNN; however the main computational bottleneck is still the ResNet encoder, which is identical across all algorithms.
> We found that RND and BYOL-Explore have nearly identical computational costs (11 learner steps/sec), and ICM is about 30% faster (14 learner steps/sec), since ICM doesn’t need to compute a second ResNet encoder for the targets.
>
> Concerning Q2, the BYOL-Explore intrinsic reward is closer to capturing the model uncertainty of predicting future observations, rather than visitation of a (o, a) pair, especially in partially observable environments. In deterministic and fully observable environments such as Atari, however, you could say that this uncertainty captured is purely the epistemic uncertainty of the state. We have observed empirically that the prediction loss becomes small enough where the model is "correct", so that adding up small losses over the open-loop steps amounts to less than the loss at a single step where the model is "incorrect".
>
> Concerning Q3, this is indeed a small bias in our current code. However in earlier experiments we did try to correct for this by using the mean, but it ended up making no difference. Because our learner is seeing subsequences of length 64 from many different parts of an episode, the effect of this bias gets greatly diffused throughout the entire episode and ends up being insignificant.

---

### Author Response · Authors · 2022-08-01
**General author feedback**

We would like to thank the reviewers for their questions and comments which allowed us to improve the paper. Here is a summary of the changes between the submitted draft and the new rebuttal draft:

 - The x axis of the figures are now expressed in number of frames instead of learner steps to allow easier comparison to prior work.
 - We made clearer the fact that we do not use sticky actions.
 - We added a discussion on stochastic distractors in Appendix A.4.3 with experiments showing robustness of BYOL-Explore and collapse of RND with a controllable noise in a noisy variant of Montezuma Revenge .
 - We added missing citations.

We also provided individual answers to each reviewer to respond to their comments and concerns.

---

### Meta-Review · Area_Chair_LsDS · 2022-08-21

**Recommendation:** Accept
**Confidence:** Certain

**Metareview:**

The reviewers agreed overall that this work is a solid contribution, combining multiple earlier methods/ideas. The experimental validation is good. Together, these merit acceptance.

**Award:**

No

---

### Decision · Program_Chairs · 2022-09-14

Accept